# Sunlight-Powered Reverse Water Gas Shift Reaction Catalysed by Plasmonic Au/TiO_2_ Nanocatalysts: Effects of Au Particle Size on the Activity and Selectivity

**DOI:** 10.3390/nano12234153

**Published:** 2022-11-23

**Authors:** Jordi Volders, Ken Elen, Arno Raes, Rajeshreddy Ninakanti, An-Sofie Kelchtermans, Francesc Sastre, An Hardy, Pegie Cool, Sammy W. Verbruggen, Pascal Buskens, Marlies K. Van Bael

**Affiliations:** 1Design and Synthesis of Inorganic Materials (DESINe), Institute for Materials Research, Hasselt University, Agoralaan Building D, 3590 Diepenbeek, Belgium; 2Imec Vzw, Imomec Associated Laboratory, Wetenschapspark 1, 3590 Diepenbeek, Belgium; 3EnergyVille, Thor Park 8320, 3600 Genk, Belgium; 4Sustainable Energy, Air & Water Technology (DuEL), University of Antwerp, Groenenborgerlaan 171, 2020 Antwerp, Belgium; 5NANOlab Center of Excellence, University of Antwerp, Groenenborgerlaan 171, 2020 Antwerp, Belgium; 6The Netherlands Organisation for Applied Scientific Research (TNO), High Tech Campus 25, 5656 AE Eindhoven, The Netherlands; 7Laboratory of Adsorption and Catalysis, University of Antwerp, Universiteitsplein 1, 2610 Wilrijk, Belgium

**Keywords:** plasmonic, nanoparticle, gold, titania, catalysis, CCU, carbon dioxide, syngas, solar fuel

## Abstract

This study reports the low temperature and low pressure conversion (up to 160 °C, *p* = 3.5 bar) of CO_2_ and H_2_ to CO using plasmonic Au/TiO_2_ nanocatalysts and mildly concentrated artificial sunlight as the sole energy source (up to 13.9 kW·m^−2^ = 13.9 suns). To distinguish between photothermal and non-thermal contributors, we investigated the impact of the Au nanoparticle size and light intensity on the activity and selectivity of the catalyst. A comparative study between P25 TiO_2_-supported Au nanocatalysts of a size of 6 nm and 16 nm displayed a 15 times higher activity for the smaller particles, which can only partially be attributed to the higher Au surface area. Other factors that may play a role are e.g., the electronic contact between Au and TiO_2_ and the ratio between plasmonic absorption and scattering. Both catalysts displayed ≥84% selectivity for CO (side product is CH_4_). Furthermore, we demonstrated that the catalytic activity of Au/TiO_2_ increases exponentially with increasing light intensity, which indicated the presence of a photothermal contributor. In dark, however, both Au/TiO_2_ catalysts solely produced CH_4_ at the same catalyst bed temperature (160 °C). We propose that the difference in selectivity is caused by the promotion of CO desorption through charge transfer of plasmon generated charges (as a non-thermal contributor).

## 1. Introduction

Current society has a very large energy demand with fossil fuels as the major energy source. This results in large-scale emissions of the greenhouse gas CO_2_ [1]. A possible method for reducing emissions is to close the carbon cycle by reutilising (CCU) the produced CO_2_. A suitable process for CCU is the reverse water gas shift (rWGS) reaction, in which CO_2_ and green H_2_ are converted to CO and water (Equation (1)) [2]. The produced CO can subsequently act as a building block for the production of long-chain hydrocarbon fuels using the Fischer–Tropsch process, thus closing the carbon cycle [3,4]. The rWGS reaction is endothermic, and conventionally requires high temperature and pressure of >700 °C and >2 MPa, respectively, to shift the equilibrium to CO production and suppress the competing CO_2_ methanation reaction (Equation (2)) [4,5,6].
(1)$${\mathrm{CO}}_{2}+H_{2}\;⇌\;\mathrm{CO}\;+{\;H}_{2}O\;\Delta H_{298K}=41.2{\;\mathrm{kJ}\;\mathrm{mol}}^{-1}$$
(2)$${\mathrm{CO}}_{2}+4H_{2}\;\to {\;\mathrm{CH}}_{4}+2H_{2}O\;\Delta H_{298K}=-165.0{\;\mathrm{kJ}\;\mathrm{mol}}^{-1}$$

These extreme conditions require a large amount of energy and could lead to sintering of catalyst material, resulting in performance losses [6,7]. To avoid sintering and to minimise the environmental impact, it is more advantageous to carry out this reaction closer to ambient conditions. A possible solution would be to use sunlight as a sustainable energy source in combination with a suitable photocatalyst [8,9,10]. Semiconductors are a well-known class of photocatalyst, of which TiO_2_ is an established example often used in applications because of its stability, cost effectiveness, and photocatalytic activity [11,12,13,14,15,16,17,18,19,20,21,22]. However, because of its large bandgap (3.2 eV, anatase) TiO_2_ only absorbs UV light which is equivalent to merely 4% of the total solar spectrum. Absorption over a broader visible light range is beneficial for maximising sunlight harvesting efficiency and to realise a high space-time-yield in sunlight-powered chemical processes. This can be achieved using plasmonic catalysts comprised of metal nanoparticles which absorb light over a broader range of the solar spectrum [23,24,25,26,27,28,29,30,31,32,33,34,35]. Plasmonic particles exhibit a localised surface plasmon resonance (LSPR) upon illumination. Au is a plasmonic metal that exhibits a strong, broad LSPR in the visible spectrum and has been shown to induce a large selectivity towards the formation of CO when applied on a semiconductive support [36]. This type of catalyst can influence catalytic chemical processes in multiple ways. First, optical near-field enhancement can occur where light is re-emitted and concentrated on the surface at the same frequency as the incident light, which is interesting for photo-activated reactions. The second contributor is local heat generation in the plasmonic particles through the Joule effect, which creates a large difference between the temperature of the catalyst particles and the reactor vessel, allowing for efficient local heating. The third effect is the generation of hot electrons in the plasmonic particles, which could then be transferred to unoccupied molecular orbitals of adsorbed reactants and initiate bond dissociation. Since the plasmonic particles are deposited on a semiconducting oxide, the hot electrons could be injected into the conduction band of the semiconductor over the Schottky barrier, increasing their lifetime and the likelihood of reactions occurring [37,38]. Finally, the band gap of the semiconductor could also be excited through UV absorption and activate the reactants. Near-field enhancement is not likely to directly contribute as the molecules involved in the conventional rWGS reaction are not light-activated. It could, however, result in additional excitation of charge carriers in the semiconductor [39]. Targeted experiments are required to differentiate between photothermal and non-thermal contributors [37,40,41,42,43,44].

Several studies have been published on the plasmon-catalysed light-driven rWGS reaction with various supported metal catalysts: Ag [45], Al [46], Cu [47], Fe [48], Pd [49,50,51,52], and Pt [52,53,54]. In addition, supported Au catalysts have been reported in the literature. Values reported as g_cat_^−1^ are normalised on the mass of both plasmonic particles and support, while values reported as g_Au_^−1^ are normalised on the mass of plasmonic particles in the catalyst. Hubert and co-workers reported Au NPs deposited on various semiconducting and dielectric metal oxides, viz., Au/TiO_2_, Au/CeO_2_, Au/Al_2_O_3_ and Au/SiO_2_, with a respective Au NP size of 3 ± 1 nm, 5 ± 1 nm, 10 ± 5 nm and 2 ± 1 nm. The catalyst materials were tested using visible light (5216 W m^−2^) at 400 °C, and a CO production rate of 159.8 mmol g_cat_^−1^ h^−1^ was observed for Au/TiO_2_, 85.02 mmol g_cat_^−1^ h^−1^ for Au/CeO_2_ and 7.08 mmol g_cat_^−1^ h^−1^ for Au/Al_2_O_3_; Au/SiO_2_ was tested with additional heating at 300 °C and a CO production rate of 11.24 mmol g_cat_^−1^ h^−1^ was observed [47,55]. Zhang et al. reported an Au/Al_2_O_3_ catalyst with an Au NP size of 2.5 ± 0.4 nm, which was tested at 350 °C under UV illumination (30,000 W m^−2^) with a CO production rate of approximately 1.3 mmol g_cat_^−1^ h^−1^ [56]. Tahir et al. investigated Au/TiO_2_ with a Au NP size of approximately 20 nm and a CO activity of 4.144 mmol g_cat_^−1^ h under 1500 W m^−2^ illumination [57]. Martinez Molina et al. reported a Au/TiO_2_ catalyst with a Au NP size of 1.6 nm, and achieved a CO production rate of 429 mmol g_Au_^−1^ h^−1^ under simulated solar light (14,400 W m^−2^) without external heating [36]. A wide range of Au NP sizes and experimental conditions has been reported, unfortunately complicating the correlation between catalytic performance and physical properties (i.e., structure and composition) of the catalyst. To date, the impact of the Au nanoparticle size on the performance of plasmonic Au/TiO_2_ catalysts in the sunlight-powered rWGS reaction has not yet been systematically investigated. A study on this subject could contribute to the identification of photothermal and non-thermal contributors to this reaction, which is of vital importance both for fundamental understanding and rational further development towards application in industrial processes.

In order to contribute to the identification of photothermal and non-thermal contributors for the sunlight-powered rWGS process, we studied and compared the catalytic performance (selectivity and activity) of Au/TiO_2_ upon illumination with artificial sunlight (irradiance up to 13.9 suns = 1.39 W·cm^−2^) and in dark. We compared the effect of small (S) (6 ± 2 nm) and medium-sized (M) Au nanoparticles (16 ± 4 nm) supported on the same TiO_2_ support material (anatase-rutile mixture, P25), and for three different Au loadings, ranging from 0.85% to 3.9% *w*/*w*. The catalysts comprising small Au nanoparticles were prepared via deposition-precipitation, and the catalysts comprising medium-sized Au nanoparticles via photo-impregnation. Chemical composition, structure and optical properties of the catalysts were studied in detail. Through this systematic study, we demonstrate that both small and medium-sized TiO_2_-supported Au nanoparticles produce CO with high selectivity using mildly concentrated sunlight as the sole source of energy at low temperature and pressure (reactor near room temperature, catalyst bed temperature ≤ 160 °C, *p* = 3.5 bar). Furthermore, we demonstrate that both thermal and non-thermal contributors play an essential role in this process, and that catalysts comprising small Au nanoparticles are much more active than their medium-sized counterparts. This activity difference is larger than expected based on the difference in total Au surface area.

## 2. Materials and Methods

TiO_2_ supported Au nanoparticles were prepared using both a deposition-precipitation method (DP) and photo-impregnation method (PI). Synthesis using the DP method was carried out as follows: urea (3.0 g, 50 mmol, VWR Chemicals, 99%) was dissolved in ultra-filtered water (Milli-Q Millipore, Billerica, MA, USA, 18.2 MΩ cm, 20 mL). TiO_2_ (490 mg, Aeroxide TiO_2_ P25, Evonik, Essen, Germany) was weighed in a 100 mL round bottom flask. The urea solution was added to the flask along with ultra-filtered water (Milli-Q Millipore, 18.2 MΩ cm, 30 mL). Next, 1.255 mL, 2.540 mL and 5.180 mL of aqueous solution of HAuCl_4_ (0.02 M, Alfa Aesar, 99.99%) were added to obtain catalysts with 1 w%, 2 w% and 4 w% Au, respectively, with a catalyst notation of S1, S2 and S4. The round bottom flask was ultrasonicated for 1 min and subsequently stirred and heated at 80 °C for 3 h. Next, freshly prepared NaBH_4_ (5 mL, 0.1 M, Acros Organics, 99%) was added and the mixture was stirred for 30 min at 80 °C. Afterwards, the catalysts were centrifuged and washed three times with water and once with ethanol. They were dried overnight in the fume hood and ground with mortar and pestle for characterisation and catalysis experiments.

Synthesis using the photo-impregnation method (PI) was performed using a modified Turkevich method [58]. Initially, an aqueous HAuCl_4_.3H_2_O solution (2.5 mL, 0.01 M, Sigma-Aldrich, St. Louis, MO, USA, >99.9%) was diluted to a total volume of 97.5 mL in a two-neck round-bottom flask, and heated to 100 °C. When the boiling point was reached, an aqueous sodium citrate (2.5 mL, 1 w%, Chem-lab) solution was added under vigorous stirring, and the reaction mixture was left under reflux for 30 min. Afterwards, the solution was rapidly cooled down to room temperature, and the Au concentration in colloidal suspension was quantitatively determined by Spectroquant analysis (NOVA 60, Merck, Rowey, NJ, USA) with a standard gold reagent test kit (114,821). Next, for the preparation of the PI catalysts with 1 w%, 2 w%, and 4 w% Au (denoted M1, M2, and M4 respectively), TiO_2_ (600 mg, Aeroxide TiO_2_ P25, Evonik) was suspended in appropriate amounts of colloidal gold solution, which were then vigorously stirred for 2 h under UV-A irradiation (Philips T5 BLB, 8 W, 365 nm). Finally, the fine catalyst powders were obtained by centrifuging the irradiated suspensions at 16,000 rpm for 15 min, decanting the supernatant, drying the catalysts overnight in a 90 °C oven in air, and grinding them into powder with a mortar and pestle.

UV-Visible diffuse reflectance spectrometry was carried out using a Cary UV-Vis-NIR spectrometer, Agilent Technologies. The sample (0.5 mg) was mixed with KBr (500 mg, Merck, 98%) and added to the powder holder. The reflectance was measured in scan mode at a scan rate of 10 nm/s, with a range between 200 nm and 800 nm. A blank KBr reference powder was used to construct a baseline (100% reflectance), zero reference point was constructed in absence of sample (0% reflectance). Powder X-ray diffraction was carried out using a Bruker D8 Discover (Cu K-α radiation, LynxEye detector). Measurements were carried out from 2θ value of 20° to 80° with a step size of 0.01° and 3 s of step time. The Au content of the catalysts was determined by inductively coupled plasma-optical emission spectrometry (ICP-OES, Perkin Elmer Optima 3300 dv simultaneous spectrometer, PerkinElmer, Waltham, MA, USA). Sample digestion was carried out in a 10 mL mixture of mineral acids (HCl (≥37%, TraceSELECT, for trace analysis, Honeywell chemicals):HNO_3_ (69.0–70.0%, J.T.Baker, for trace metal analysis):HF (40%, AnalaR NORMAPUR^®^ analytical reagent, VWR chemicals) in a 3:1:1 ratio) in a Milestone microwave setup. The catalysts solutions and 1000 ppm Au standard (Merck) were diluted by 5% HNO_3_ to 1–10 ppm and 10, 5, 2, 1 ppm concentrations, respectively for ICP-OES measurements. All ICP analyses were carried out *in duplo*. Scanning transmission electron microscopy (STEM) was performed using high angle annular dark field (HAADF) detector and EDX was performed using a super-EDX detector on the FEI Tecnai Osiris Microscope, operated at 200 kV. Diffraction patterns were also measured using the FEI Tecnai Osiris Microscope, operated at 200 kV. More than 100 nanoparticles were analysed per catalyst to determine particle size distribution. High resolution scanning transmission electron microscopy (HR-STEM) was performed using an aberration-corrected cubed Thermo Fischer Scientific Titan Transmission Electron Microscope, operated at 300 kV. SEM-EDX was performed with a Zeiss 450 Gemini 2 FEG-SEM equipped with a ThermoFischer EDX detection system, operated at 20 kV. Averages of four different locations are reported.

The catalytic CO_2_ hydrogenation tests were performed in a custom-made batch photoreactor, equipped with a quartz window at the top to allow for light irradiation, as described by Sastre et al. [26]. A schematic representation of this photoreactor is shown in Appendix A. Light for the photoreactions was provided by a solar simulator (Newport Sol3A) placed above the reactor and was equipped with a high flux beam concentrator (Newport 81,030), and AM1.5 filter. An area of about 3.14 cm^2^, which was fully covered by the catalyst sample, was irradiated. The reactor is equipped with three thermocouples to measure the temperatures at the top and bottom of the reactor, and at the bottom of the catalyst bed. In a typical run, 200 mg of the catalyst was put in the reactor. The reactor was made air-free after three cycles of N_2_ fill and vacuum purge cycles. Afterwards, the reactor was filled with a mixture of H_2_ (Linde 6.0), CO_2_ (Linde 4.5) and N_2_ (Linde 5.0), with a H_2_:CO_2_:N_2_ ratio of 2:2:1, until reaching a pressure of 3.6 bar. The reaction start time was considered to be when the light was switched on. For experiments in dark conditions, the reactor was heated until the desired temperature. Once the temperature stabilised, the gas mixture was introduced, which marked the start of the reaction time. In this case, no gas sample could be taken at the reaction starting time as the pressure in the vessel was too low. A sample was taken immediately when the reactor vessel pressure allowed for it. Products were analysed by a gas chromatograph (Compact GC, Global Analyzer Solutions), gas samples were taken from the reactor using a gas tight syringe at different reaction times, and directly injected in the GC. The GC was equipped with three channels, two microthermal conductivity detectors (TCD) and one flame ionisation detector (FID). The peak areas were used to determine the ratio of each compound based on calibration, using N_2_ as an internal standard. If products were present in the time zero analysis, this value was subtracted from the following results.

Flow experiments were carried out in a similar fashion as described above. Instead of filling the reactor, a constant flow of CO_2_:H_2_:N_2_ (8 mL min^−1^:8 mL min^−1^:4 mL min^−1^) was allowed through the catalyst bed and the reactor, and 3.5 bar reactor pressure was maintained with a backpressure regulator. GC injections were automatically taken every 2.5 min from the outgoing flow.

## 3. Results & Discussion

### 3.1. Catalyst Preparation and Characterisation

Au/TiO_2_ catalysts (1 w%, 2 w%, 4 w% theoretical Au loading) were synthesised via deposition-precipitation (DP) and photo-impregnation (PI) as described in the experimental section. In short, for the DP method, catalysts were prepared by mixing HAuCl_4_ and TiO_2_ (P25) with aqueous urea and heating it to 80 °C. Subsequently, NaBH_4_ was added to reduce Au^3+^ to Au. The catalysts with 1 w%, 2 w% and 4 w% theoretical Au loading produced by DP are designated S1, S2 and S4, respectively. For the PI method, the catalyst synthesis was started by heating an aqueous HAuCl_4_.3H_2_O precursor solution to 100 °C, and adding sodium citrate at the boiling point as a stabilising and reducing agent. Afterwards, the resulting Au nanoparticles were mixed with TiO_2_ (P25) and irradiated with UV-A light for photo-impregnation. Catalysts synthesised according to the PI method with theoretical Au loading of 1 w%, 2 w% and 4 w% were identified as M1, M2 and M4, respectively. The Au content was determined using inductively coupled plasma-atomic emission spectroscopy (ICP-AES). ICP-AES analyses were carried out *in duplo* for all catalysts, and a 98% confidence interval is reported. For the S catalysts, an Au weight loading of 0.85 ± 0.03% (S1), 1.85 ± 0.03% (S2) and 3.9 ± 0.5% (S4) was measured. This means that 85–97% of the nominal Au content was deposited on TiO_2_. For the PI catalysts, the Au weight loading was 0.98 ± 0.01% (M1), 1.90 ± 0.05% (M2) and 3.72 ± 0.06% (M4), which means that 93–98% of the nominal amount of Au was deposited on TiO_2_. These values are close to the expected theoretical values and will be applied for normalising the catalytic activity. High angle annular dark field-scanning transmission electron microscopy (HAADF-STEM) was carried out to analyse the size of Au particles of the catalysts (Figure 1a,b and Appendix A). High-resolution scanning transmission electron microscopy (HR-STEM) was carried out to investigate crystal plane alignment of Au and TiO_2_ particles (Appendix A). No preferential orientation of the Au lattice compared to the TiO_2_ lattice was observed. Diffraction patterns have been added in Appendix A in order to identify anatase and rutile phases in the electron microscopy images. Only (101) for anatase and (110) for rutile could be identified due to closely related d-spacing for both phases. Energy dispersive X-ray (EDX) analysis was used to identify the relative positioning of Au nanoparticles and TiO_2_ support particles (Appendix A). The Au particles are randomly deposited on the TiO_2_ support with an Au particle size of 5.5 ± 0.4 nm for S1, 5.9 ± 0.4 nm for S2 catalysts and 7.8 ± 0.8 nm for S4 catalysts, while the Au particle size of the M1 and M2 catalysts was 16 ± 1 nm and for the M4 catalyst 14.2 ± 0.5 nm (Figure 1a,b and Appendix A). This distinct difference in Au particle size will allow for determination of size-dependent effects in the catalytic system. EDX analysis confirms the identification of Au and TiO_2_ particles as major constituents of the catalysts, as expected (Appendix A). SEM-EDX (Appendix A and Appendix A) was carried out on the S4 and M4 catalysts to confirm chemical composition, and for both samples only O, Ti and Au were detected. For S4, a composition of 36 ± 6% O, 59 ± 6% Ti, and 4 ± 1% Au was found. For M4, a composition of 39 ± 5% O, 59 ± 4% Ti, and 3 ± 1% Au was found. This is in good agreement with the Au loading determined by ICP-AES. Due to the relatively large error on Au loading values determined by SEM-EDX, no significant difference between both samples could be detected. Diffuse reflectance UV-vis spectroscopy was carried out on the Au/TiO_2_ catalysts as well as on bare TiO_2_ (Figure 1c and Appendix A). In all samples, an absorption band edge in the near UV region (<400 nm) can be observed, which can be attributed to the band gap of TiO_2_. For the Au/TiO_2_ catalysts, a broad absorption band with a maximum at 550 nm is observed, which can be attributed to the LSPR of the Au nanoparticles. The position of the absorption band maximum has been shown for the S2 and M2 catalysts (Appendix A), by making use of the first derivative. No significant differences between the LSPR of S and M catalysts related to Au particle size could be observed. X-ray diffraction (XRD) was carried out on the S4 and M4 catalysts to confirm phase purity of the TiO_2_ support, and confirm the presence of metallic Au (Figure 1d). The diffraction pattern of anatase (JCPDS #21-1272) is clearly present, namely the hkl values of (101), (103), (004), (112), (200), (105), (211), (213), (204), (116), (220), (107), (215) and (301) could be correlated to 2θ values of 25.28°, 36.95°, 37.80°, 38.58°, 48.05°, 53.89°, 55.06°, 62.12°, 62.69°, 68.76°, 70.31°, 74.03°, 75.03° and 76.02°, respectively [59,60]. The diffraction pattern of rutile (JCPDS #21-1276) is also clearly present, namely the hkl values of (110), (101), (200), (111), (210), (211), (220), (002), (310), (301), (112) and (311) could be correlated to 2θ values of 27.45°, 36.09°, 39.19°, 41.23°, 44.05°, 54.32°, 56.64°, 62.74°, 64.04°, 69.01°, 69.79° and 72.41°, respectively [60,61]. Some hkl values are not visible and likely under the detection limit, namely (221), (320), (202) and (212). Characteristic Au (JCPDS #04-0784) diffraction peaks could also be observed, namely the hkl values of (111), (200), (220) and (311) could be attributed to 2θ values of 38.19°, 44.39°, 64.58° and 77.55°, respectively [60,62]. The Scherrer equation was used to estimate crystallite size of the compounds present (Appendix A), the values obtained for both anatase (20.5–21.3 nm) and rutile (29.9–33.6 nm) are in line with observations in HAADF-STEM. The crystallite size of the Au NPs could not be reliably determined due to the low intensity of the corresponding diffraction peaks. A slightly higher peak intensity of Au diffraction peaks can be observed for the M4 catalyst compared to the S4 catalysts, which can be attributed to the larger particle size causing less peak broadening.

### 3.2. Catalytic Performance

Sunlight-powered CO_2_ hydrogenation using plasmonic Au/TiO_2_ nanocatalysts was studied using a solar simulator with an artificial sunlight intensity of 13.9 kW·m^−2^ (13.9 suns, AM 1.5) for 2 h at a reactor temperature of 20 °C, without additional external heating. These catalysis experiments were carried out in a batch photoreactor with a quartz window to allow light transmittance of light from the solar simulator onto the catalyst (irradiance spectrum available, Appendix A). The reactor was filled with a mixture of 2:2:1 CO_2_:H_2_:N_2_ ratio until a pressure of 3.5 bar was reached. In dark experiments the reactor was externally heated via the bottom and sides of the reactor vessel. The catalyst bed temperature was measured by a thermocouple at the bottom of the catalyst bed (Appendix A). When the light was switched on, the temperature of the catalyst bed rapidly increased from 20 °C to 145 °C for all photocatalysts (Figure 2d) through photothermal heating. The catalyst bed temperature increased further to 160 °C during the two hour reaction time. Note: because of the single side illumination and low thermal conductivity of the catalyst bed, a temperature gradient inside the catalyst bed (in z-direction) can be expected during the reaction. Based on previously reported studies, it is reasonable to assume that at 13.9 suns irradiance the top surface temperature is about 100 °C higher than the temperature measured underneath the catalyst bed with a thermocouple [63,64,65,66,67]. The reactor temperature, calculated as the average of the temperatures at both the top and bottom of the reactor vessel, remained close to room temperature (<30 °C) for the duration of the experiment. No products besides CO and CH_4_ were observed for all reactions.

Under light conditions, the blank TiO_2_ sample yielded no product after two hours of reaction time (catalyst bed temperature = 63 °C). S1, S2 and S4 produced CO (Figure 2a–c) at an initial rate, calculated for the first 60 min reaction time, of 6.56 mmol g_Au_^−1^ h^−1^ (0.056 mmol g_cat_^−1^ h^−1^), 7.81 mmol g_Au_^−1^ h^−1^ (0.144 mmol g_cat_^−1^ h^−1^) and 6.91 mmol g_Au_^−1^ h^−1^ (0.269 mmol g_cat_^−1^ h^−1^), respectively, with a high selectivity (≥88%). The only side product formed is CH_4_, which reaches a plateau value within the first minutes of reaction (Figure 2b). The CO production increased linearly in time, and did not reach equilibrium in the two hours of reaction time under the applied conditions. Under the same reaction conditions, M1 produced only a small amount of CH_4_, and no CO could be detected. M2 and M4 produced CO with a selectivity of ≥84%, however at a much lower activity than S2 and S4, viz. 0.53 mmol CO g_Au_^−1^ h^−1^ (0.010 mmol CO g_cat_^−1^ h^−1^) and 0.44 mmol CO g_Au_^−1^ h^−1^ (0.017 mmol CO g_cat_^−1^ h^−1^), respectively. This 15-fold difference in activity could in part be attributed to the lower Au surface area of M2 and M4 as a result of its larger particle size when compared to S2 and S4, viz. 16 nm (M) vs. 6 nm Au (S) (total Au surface area S/M = 2.67). Other factors, e.g., the electronic contact between Au and TiO_2_ and the ratio between plasmonic absorption and scattering, may also contribute to this difference.

The S4 and M4 samples were studied under illumination using the photoreactor in continuous flow mode (with H_2_, CO_2_ and N_2_ flow rates of 8 mL min^−1^, 8 mL min^−1^ and 4 mL min^−1^, respectively), with varied light intensity (between 6 and 13 suns) to distinguish between photothermal and photochemical contributors (Figure 3). For a photothermal process, an exponential increase in CO production is expected according to the Arrhenius equation, as the temperature obtained by photothermal heating is linearly dependent on the light intensity. For the S4 catalyst, this exponential relationship between the activity and the applied light intensity is observed, demonstrating that a photothermal effect is indeed contributing to the rWGS reaction (Figure 3a). This is in accordance with previous studies [36,37]. For the M4 catalyst we expect that the observed relationship is also exponential (Figure 3b); however, for lower light intensities, the CO production is below the detection limit. The apparent activation energy of the rWGS reaction was not calculated with these data, as this would require accurate online monitoring of the top surface temperature and the temperature gradient inside the catalyst bed during the reaction [63]. Using only the temperature measured underneath the catalyst bed with a thermocouple does not generate adequate results.

The catalytic performance of Au/TiO_2_ catalysts was also studied in dark to elucidate the differences in catalytic performance between light and dark (Figure 4a,b). For the thermal reference experiments in dark, the reactor vessel was heated to a setpoint temperature of 168 °C. This allows for a catalyst bed temperature during reaction of around 160 °C, equivalent to the measured catalyst bed temperature after two hours of reaction under illumination (13.9 suns, *vide supra*). Only CH_4_ was produced during these dark reactions. This difference in selectivity between light and dark is a strong indication of the contribution of a non-thermal effect to the reaction upon illumination. The rate-determining step for CO production is the desorption of CO from the catalyst, while for methanation this is the dissociation of the C-O bond, followed by hydrogenation to CH_4_ [56]. We therefore hypothesise that the difference in catalyst selectivity was caused by promotion of CO desorption from the catalyst surface through charge transfer of plasmon generated charges, which is in line with what has been previously reported in the literature [36,37,47]. All Au/TiO_2_ catalysts yield a CH_4_ production rate in the range of 0.52–1.51 mmol g_cat_^−1^ h^−1^, which is in the same order of magnitude as the production rate obtained with bare P25 TiO_2_ without Au (0.92 mmol g_cat_^−1^ h^−1^). This indicates that Au does not contribute to CH_4_ formation in the dark and that TiO_2_ is the actual catalyst in this process. To take the underestimation of catalyst bed temperature into account, verification experiments were carried out with a reactor vessel temperature setpoint at its maximum of 208 °C (resulting in a catalyst bed temperature of 200 °C, Appendix A). A CH_4_ production rate in the range of 2.64–8.87 mmol g_cat_^−1^ h^−1^ was achieved, and no CO was produced. The increased activity for CH_4_ production is expected with a higher reaction temperature. In previously reported Au/TiO_2_ work, a shift in selectivity towards CH_4_ was also observed, but significant CO production was also still detected [36]. A possible reason for this difference could be the use of a different support material phase (pure anatase) and the difference in Au particle size.

## 4. Conclusions

We reported on the synthesis of plasmonic Au/TiO_2_ catalysts comprising small (S, 6 nm) and medium-sized (M, 16 nm) Au nanoparticles using a deposition-precipitation method with chemical reduction and photo-impregnation, respectively. The chemical composition, nanostructure and optical properties of these catalysts were determined. Under mild reaction conditions (reactor vessel close to room temperature, 3.5 bar pressure) using slightly concentrated artificial sunlight (13.9 suns), the S Au/TiO_2_ photocatalysts yielded a higher activity (7.81 mmol CO g_Au_^−1^ h^−1^) and similar selectivity (≥88%) compared to the M Au/TiO_2_ photocatalysts (0.53 mmol CO g_Au_^−1^ h^−1^; ≥84% selectivity). The 15-fold difference in activity can in part be attributed to smaller Au surface area in M photocatalysts compared to S photocatalysts. A photothermal contribution to the rWGS reaction was demonstrated for both S and M photocatalysts by identifying an exponential relationship between catalytic activity and light intensity. In dark conditions, all catalysts produce solely CH_4_ instead of CO at a rate in the range of 0.52–1.51 mmol g_cat_^−1^ h^−1^, which is in the same order of magnitude as the production rate obtained with bare P25 TiO_2_ without Au (0.92 mmol g_cat_^−1^ h^−1^). This indicates that Au does not significantly contribute to CH_4_ formation in dark, and that TiO_2_ is the actual catalyst in this process. This large selectivity difference between light and dark strongly indicates the presence of a non-thermal contributor to the process. This non-thermal contribution may encompass promotion of CO desorption from the catalyst surface through charge transfer of plasmon-generated charges.

## Figures and Tables

**Figure 1 nanomaterials-12-04153-f001:**
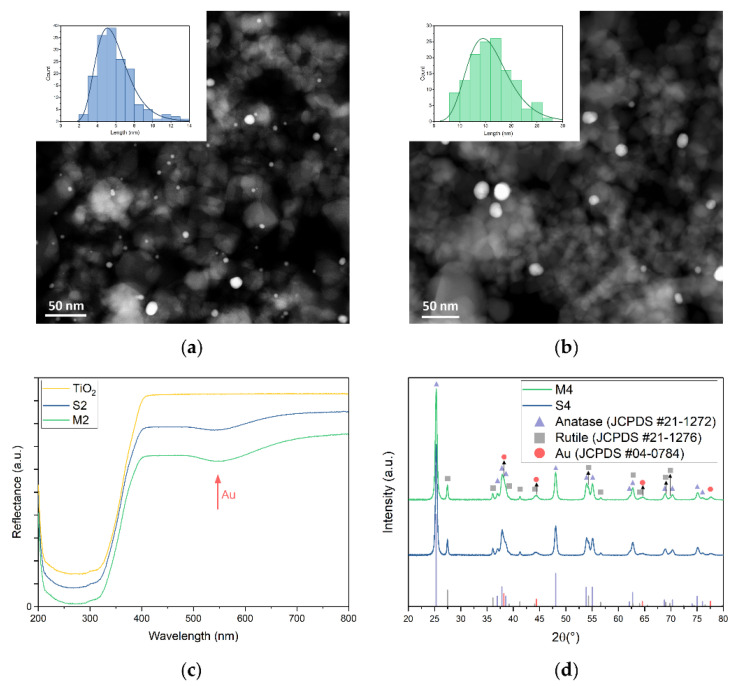
(**a**) HAADF-STEM image of S2 catalyst with Au nanoparticle size histogram, consisting of a count (bars) and lognormal plot (curve) of particle sizes; (**b**) HAADF-STEM image of M2 catalyst with Au nanoparticle size histogram, consisting of a count (bars) and lognormal plot (curve) of particle sizes; (**c**) diffuse reflectance UV-vis spectra of bare P25 TiO_2_, S2 and M2 catalysts. The plasmonic Au absorption band is highlighted; (**d**) X-ray diffractograms of S4 and M4 catalysts. Anatase, rutile and metallic Au are identified using PDF database [59,60,61,62].

**Figure 2 nanomaterials-12-04153-f002:**
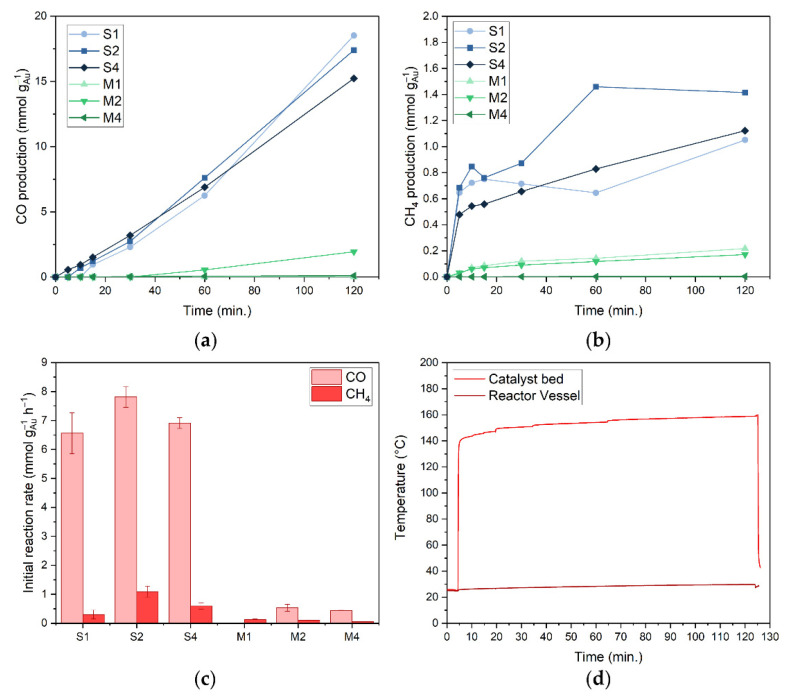
(**a**) CO production of S1–4 and M1–4 catalysts under illumination; (**b**) CH_4_ production of S1–4 and M1–4 catalysts under illumination; (**c**) initial reaction rates of S1–4 and M1–4 catalysts for both CH_4_ and CO production under illumination. Error bars represent a 98% confidence interval; (**d**) catalyst bed and reactor vessel temperature of S2 catalyst during reaction. Connecting lines are a guide to the eye. Reaction conditions: mixture of CO_2_:H_2_:N_2_ (2:2:1) at 3.5 bar pressure, 200 mg Au/TiO_2_ catalyst, 13.9 sun irradiation from solar simulator (AM1.5); 1 sun = 1 kW m^−2^.

**Figure 3 nanomaterials-12-04153-f003:**
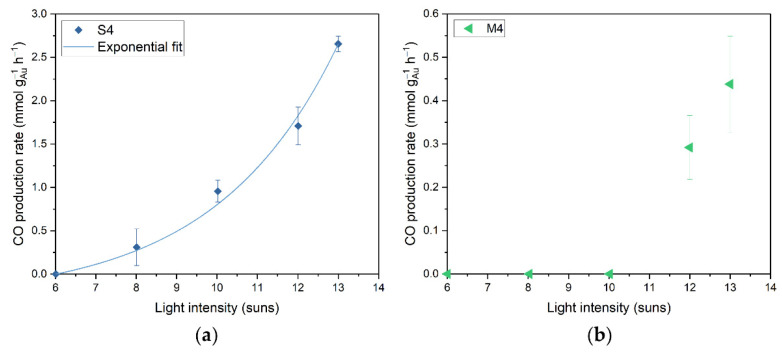
(**a**) CO production rate as function of light intensity for the S4 catalyst. The curve is an exponential fit for the S4 dataset; (**b**) CO production rate as function of light intensity for the M4 catalyst. Error bars represent a 98% confidence interval. Reaction conditions: flow process with a mixture of CO_2_:H_2_:N_2_ (8 mL min^−1^:8 mL min^−1^:4 mL min^−1^) at 3.5 bar pressure, 200 mg catalyst, varied intensity irradiation from solar simulator (AM1.5); 1 sun = 1 kW m^−2^.

**Figure 4 nanomaterials-12-04153-f004:**
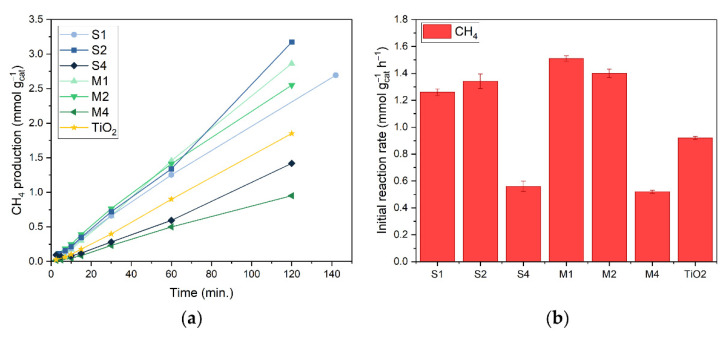
(**a**) CH4 production of bare TiO2, M1–4 and S1–4 catalysts in the dark per gram of catalyst powder; (**b**) initial reaction rates of bare TiO2, S1–4 and M1–4 catalysts for CH4 production in dark while no CO production was observed. Error bars represent a 98% confidence interval. Connecting lines are a guide to the eye. Reaction conditions: mixture of CO2:H2:N2 (2:2:1) at 3.5 bar pressure, 200 mg catalyst, dark conditions (160 °C catalyst temperature).

## Data Availability

Not applicable.

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
