# Peer review of "Sunlight-Powered Reverse Water Gas Shift Reaction Catalysed by Plasmonic Au/TiO2 Nanocatalysts: Effects of Au Particle Size on the Activity and Selectivity"

_nanomaterials, 2022, doi:10.3390/nano12234153_

Round 1

Reviewer 1 Report

Dear Editor: I would like to express my deep thanks for inviting me to review the manuscript ID: nanomaterials-2001802-peer-

Title:      Sunlight-powered reverse water gas shift reaction catalyzed by plasmonic Au/TiO2 nanocatalysts: effects of Au particle size on the activity and selectivity

Authors: Jordi Volders, Ken Elen, Arno Raes, Rajeshreddy Ninakanti, An-Sofie Kelchtermans, Francesc Sastre, An Hardy, Pegie Cool, Sammy W. Verbruggen, Pascal Buskens, Marlies K. Van Bael

Comments:

Please submit the manuscript according to Nanomaterials template.

Introduction:

The first paragraph of introduction part is not clear. Need to rewrite introduction part. For example, importance of this work, explain the synthesis process the objectives and novelty and so on. Follow below references and cited them

1.         M. Zeshan, I. A. Bhatti, M. Mohsin, M. Iqbal, N. Amjed, N. AlMasoud,T.S. AlomaRemediation of pesticides using TiO2 based photocatalytic strategies: A review” Chemosphere 300 (2022) 134525

2.         B.T. Lee, J.K. Han, A.K. Gain, K.H. Lee, F. Saito, “TEM microstructure characterization of nano TiO2 coated on nano ZrO2 powders and their photocatalytic activity” Materials Letters 60 (17-18), (2006) 2101-2104.

3.         M.A.E. Wafi, M.A. Ahmed, H. S. Abdel-Samad, H.A.A Medien, “Exceptional removal of methylene blue and p-aminophenol dye over novel TiO2/RGO nanocomposites by tandem adsorption-photocatalytic processes” Materials Science for Energy Technologies 5 (2022) 217-231

Results and discussion:

1.             Please provide HRTEM image with diffraction pattern to differentiate the TiO2 phase in Figure 1. It is important to provide point EDS data in Figure 1 that can provide phase identification. Please read this paper, Materials Letters 60 (17-18), (2006) 2101-2104.

2.             The Xrd analysis data is not clear. Redraw the profiles and search all peaks and marked it.

3.             Please add all samples data in Figure 3 (a) and (b) and explain the mechanism.

4.             Provide the error bar in Figure 4 (b).

Experimental

“Experimental” replaced by “Experimental procedures”

Explain in detail the characterization section

Conclusion part:

Please rewrite the conclusion part in bullet points.

RECOMMENDATION

After reviewing the enclosed manuscript for “Nanomaterials”, the present manuscript contains some kinds of scientific analysis but it is mandatory required to modify according to the preceding remarks. So, the manuscript can be publication after major revision.

Reviewer 2 Report

Report on the manuscript nanomaterials-2001802 entitled “Sunlight-powered reverse water gas shift reaction catalyzed by plasmonic Au/TiO2 nanocatalysts: effects of Au particle size on the activity and selectivity”.

The submitted manuscript should be revised. The following points should be addressed

1.  The language of the manuscript should be revised.

2. Figure 1 title is away from the figure. The title of the figure should be after the figure. Please, organize the figures and their titles. DLS should be studied to confirm the particles size.

3. In XRD analysis, the JCPDS should be indicated in the discussion and the crystal size should be estimated for all studied materials. Recommended reference for JCPDS card No. 00-021-1272 is [Applied Surface Science, Volume 400, 2017, Pages 355-364] and the reference of rutile JCPDS Card # 89–8304 is [Chemosphere, 293, April 2022, 133540]. The JCPDS of Au should be revised and found the suitable reference.

4. The chemical content of the prepared Titania should be studied via EDX and XPS.

5. What about the regeneration of the prepared nanocatalysts after the first cycle of catalysis.

6. The conclusion part should be more concise and scientific including the main achievement.

Round 2

Reviewer 1 Report

Authors addressed all the review comments in the revived manuscript. 

Reviewer 2 Report

The revised version could be accepted